# FACS: FAST ADAPTIVE CHANNEL SQUEEZING

## ABSTRACT

Channel squeezing is one of the central operations performed in CNN bottlenecks to reduce the number of channels in a feature map. This operation is carried out by using a $1 \times 1$ pointwise convolution which constitutes a significant amount of computations and parameters in a given network. ResNet-50 for instance, consists of 16 such layers which form 33% of total layers and 25% (1.05B/4.12B) of total FLOPs or computations. In the light of their predominance, we propose a novel "Fast Adaptive Channel Squeezing" module which carries out the squeezing operation in a computationally efficient manner. The key benefit of FACS is that it neither alters the number of parameters nor affects the accuracy of a given network. When plugged into diverse CNNs architectures, namely ResNet, VGG, and MobileNet-v2, FACS achieves state-of-the-art performance on ImageNet and CIFAR datasets at dramatically reduced FLOPs. FACS also cuts the training time significantly, and lowers the latency which is particularly advantageous for fast inference on edge devices. The source-code will be made publicly available.

## 1    INTRODUCTION

Introduced by ResNet (He et al., 2016), squeeze-and-expand units form the basis of state-of-the-art CNNs. It is essentially a three layered unit in which the first layer ($1 \times 1$) performs channel squeezing while the third layer ($1 \times 1$) performs channel expansion. The middle layer ($3 \times 3$), on the other hand maintains the channel count, and governs the network's receptive field. Interestingly, in CNNs inheriting squeeze-and-expand units, we make a key structural observation; that the squeeze and expand layers dominate both in the number and computations, while do not contribute to a network's receptive field due to their pointwise nature. For instance, ResNet-50 consists of 32 such layers out of 50 total layers, accounting for $\sim 54\%$ (2.23B/4.12B) of overall FLOPs, whereas ResNet-101 consists of 66 of them out of 101, accounting for $\sim 50\%$ (3.98B/7.85B) of total FLOPs.

As CNNs are widely used in machine vision (Ren et al., 2015; Zhao et al., 2017; Carion et al., 2020), bigger networks are now preferred to achieve higher accuracy Gao et al. (2018) to deal with increased task complexity. For this reason, VGG, ResNet style networks are still dominant in both academia and industry (Kumar & Behera, 2019; Ding et al., 2021; Kumar et al., 2020) due to their architectural simplicity, customizability, and high representation power, in contrast to newer complex networks (Tan & Le, 2019). However, as ResNet-like CNNs are based on squeeze-expand units, our preceding observation raises an important question: "*can computations in $1 \times 1$ layers be reduced without sacrificing network parameters and accuracy?*" If so, the inference of such networks can be significantly accelerated on edge computing devices, benefiting a whole spectrum of applications such as autonomous driving, autonomous robotics, and so on.

To the best of our knowledge, the above problem has not been addressed previously but is of great practical importance. Hence, in this paper, we show that indeed it is possible to achieve the desired objective by examining channel squeezing through the lens of computational complexity. To this end, we propose a novel "fast adaptive channel squeezing" (FACS) module that transforms a feature map $\mathbf{X} \in \mathbb{R}^{C \times H \times W}$ into another map $\mathbf{Y} \in \mathbb{R}^{\frac{C}{\mathcal{R}} \times H \times W}$, mimicking the functionality of a squeeze layer but with fewer computations, while retaining a network's parameters, accuracy, and non-linearity.

We evaluate FACS by embedding it into three CNN architectures: plain (VGG), residual (ResNet), and mobile-series (MobileNet-v2), and on three datasets: ImageNet, (Deng et al., 2009), and CIFAR-10, CIFAR-100. FACS is backed with comprehensive ablation study. Since FACS is novel, we demonstrate intermediate data representations learnt by FACS using GradCAM (Selvaraju et al., 2017), and show that they are better relative to the baselines. FACS brings huge improvements e.g. ResNet-50 becomes $\sim 23\%$ faster with 0.47% improved accuracy on ImageNet (Deng et al., 2009), whereas VGG becomes faster by $\sim 6\%$ with 0.23% improved accuracy, without bells and whistles.

Next section deals with the related work. Sec. 3 describes the FACS module, and its integration into different CNNs. Sec. 4 carries solid experiments in support of FACS. Sec. 5, concludes our findings.

## 2 RELATED WORK

### 2.1 CONVNETS

The earlier CNNs (Krizhevsky et al., 2012; Simonyan & Zisserman, 2014; He et al., 2016) are accuracy oriented but have grown complex in terms of branching (Huang et al., 2017), despite having higher accuracy. Mobile networks (Sandler et al., 2018; Zhang et al., 2018) focus on lower FLOPs for high speed inference by using depthwise convolutions (Sifre & Mallat). However they suffer from low representation power, need prolonged training schedule of $200 - 400$ epochs in contrast to earlier ones which are typically trained for $90 - 120$ epochs. The limited representation power hinders their performance on downstream tasks e.g. (Howard et al., 2017) needs 200 epochs on ImageNet to perform similar to (Simonyan & Zisserman, 2014) (trained for 75 epochs), but performs poorly on object detection. (Tan & Le, 2019) tackles this issue by being efficient in parameters and FLOPs, however ends-up being highly branched and deeper, memory hungry, and sluggish (Ding et al., 2021). Recent Transformer based methods are computationally hungry due to their attention mechanisms (Vaswani et al., 2017; Dosovitskiy et al., 2020) which put them out of the reach of edge-devices.

ResNet design space exploration (Schneider et al., 2017) provides several variants competitive to (Tan & Le, 2019) like designs while being simpler, quite advantageous for edge devices and real-time applications. Similarly, (Ding et al., 2021) improves the speeds of VGG however its training time design has many branches which is still inferior to ResNet. Furthermore, (Hu et al., 2018; Woo et al., 2018) propose novel units which yield improved accuracy in ResNet at the expense of increased parameters because of additional convolutions, and marginal computational overhead. The above discussion shows that the older architectures still have a room for improvement, and real-world applications can be benefited by focusing on these areas.

### 2.2 FAST INFERENCE

Fast inference of CNNs has been widely explored which mainly includes static pruning, network compression / quantization. These methods are generally employed post training and are agnostic to any given network. More recently, dynamic pruning methods such as (Gao et al., 2018) have become state-of-the-art in this area, which deals with the limitation of static pruning methods. Pruning works mainly by disabling or suppressing a set of channels in a feature maps, and for which convolution computations are inhibited. For example, (Gao et al., 2018) consumes a tensor $X \in \mathbb{R}^{C \times H \times W}$ and outputs another tensor $Y \in \mathbb{R}^{C \times H \times W}$, but some of the channels in $\mathbf{Y}$ have zero values for which convolution operations are inhibited, thus saving computations but at the cost of accuracy. (Gao et al., 2018) achieves its goal by introducing new convolution layers into a network.

The proposed FACS is orthogonal to the above methods since it performs computationally efficient channel squeezing, transforming a tensor $X \in \mathbb{R}^{C \times H \times W}$ to $Y \in \mathbb{R}^{\frac{C}{R} \times H \times W}$. In addition, FACS preserves a network's parameters and accuracy, and does not involve new convolution layers, rather reuses the parameters of the original squeeze layer while redefining the information flow. Hence, FACS and pruning can not complement each other. However as pruning is applicable to any network architecture, and as FACS is an architectural enhancement, thus it can be seamlessly integrated with any pruning method to speed-up the overall network. However, in this paper, we have limited ourself to the study of FACS in deep networks.

## 3 FACS

Given an input feature map $\mathbf{X} \in \mathbb{R}^{C \times H \times W}$, FACS progressively infers intermediate 1D descriptors $\mathbf{z} \in \mathbb{R}^{C \times 1 \times 1}$, $\mathbf{p} \in \mathbb{R}^{\frac{C}{R} \times 1 \times 1}$, and an output feature map $\mathbf{Y} \in \mathbb{R}^{\frac{C}{R} \times H \times W}$ of reduced dimensionality. FACS can primarily be sectioned into three stages: First, global context aggregation which provides channel-wise global spatial context in form of a 1D descriptor $\mathbf{z}$. Second, cross channel information blending which transforms $\mathbf{z}$ into another descriptor $\mathbf{p}$, referred as adaptive channel fusion probability. Third, channel fusion which utilizes $\mathbf{p}$ and $\mathbf{X}$ in order to produce $\mathbf{Y}$. The overall structure of FACS module is illustrated in Figure 2.

For reference, we have also shown the baseline channel squeezing method in Figure 1. It can be noticed that FACS is substantially different from the baseline counterpart in term of structure. The only component shared between them is the convolution layer, yet the tensors on which it operates are different. Further, FACS commonly employs global pooling, maximum or averaging operations which are the most fundamental operation in neural networks, however their use in FACS configuration is entirely novel. For this reason, we term FACS as a novel module to perform fast channel squeezing.

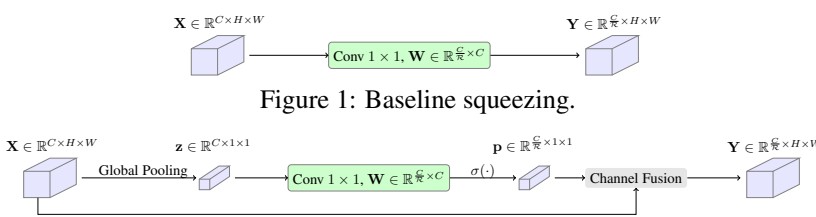

Figure 1: Baseline squeezing.

Figure 2: FACS block. $\sigma(\cdot)$ denotes sigmoid non-linearity.

### 3.1 GLOBAL CONTEXT AGGREGATION

The first and foremost step of FACS is to gather global spatial context. For this purpose, we employ global average pooling over the per-channel activations of the input feature map $\mathbf{X}$. We apply the operator on $\mathbf{X}$ which results in channel-wise global contextual information in form of a 1D descriptor $\mathbf{z} \in \mathbb{R}^{C \times 1 \times 1}$. In the computations, each element $z_i$ corresponding to $i^{th}$ channel of $\mathbf{z}$ is given by:

$$z_i = \frac{1}{H \times W} \sum_{h \in H, w \in W} X_i(h, w) \tag{1}$$

The use of global pooling in our case is inspired by its success to aggregate global statistics in a computationally efficient manner e.g. at the end of the classification networks (He et al., 2016; Hu et al., 2018; Woo et al., 2018)

### 3.2 CROSS CHANNEL INFORMATION BLENDING

From the previous step, we obtain global context which is limited to a single channel only, therefore the descriptor $\mathbf{z}$ is devoid of cross-channel information. To mitigate the issue, the per-channel global information in $\mathbf{z}$ is now blended with the other channels by using a $1 \times 1$ convolution operator. In addition to the blending, the operator also reduces the input number of channels by a factor of $\mathcal{R}$ i.e. number of input channels are reduced from $C$ to $C/\mathcal{R}$. It is important to note that the convolution operator is same as the squeeze layer in Figure 1, but operates only upon a 1D descriptor instead of a 3D tensor. This is where the majority of computation savings occur in FACS relative to the baseline.

Next, the output of the convolution operator is squeezed into a range $\in [0, 1]$ using sigmoidal activation, resulting in a 1D descriptor $\mathbf{p} \in \mathbb{R}^{C/\mathcal{R}}$. We refer the descriptor $\mathbf{p}$ as adaptive channel fusion probability which accommodates both the spatial, and the channel context. Now both the $\mathbf{p}$ and $\mathbf{X}$ are forwarded to the channel fusion stage which yield the final output $\mathbf{Y}$ of the FACS module.

### 3.3 CHANNEL FUSION

In this stage, the task of channel squeezing of $\mathbf{X}$ is performed in an adaptive and computationally efficient manner. We define two reduction operations: maximum (Max), and average (Avg). During fusion, the input feature map is first virtually partitioned into chunks of $\mathcal{R}$ channels, and then the channels of each chunk are adaptively fused to produce a single channel. In other words, fusion operation results in $C/\mathcal{R}$ output channels, if the number of input channels were $C$. We devise two strategies to perform this task, as discussed below.

#### 3.3.1 THRESHOLD BASED FUSION

In this strategy, the channels are fused by applying the reduction operator (Max or Avg) at each of the spatial location $\in H \times W$ across $\mathcal{R}$ channels. For any spatial location $(h, w) \in H \times W$, there exist $\mathcal{R}$ scalar values which are reduced to a single scalar by using one of the reduction operator. The selection of the reduction operator is entirely adaptive because it is determined by the fusion probability $\mathbf{p}$, which in turn is learnt by the network itself during the training. In short, as soon as the reduction operator is selected, the values at any given location are fused via the following equations:

$$Y_i(h, w) = \begin{cases} p_i * \text{Max}(X_j(h, w), ..X_{j+\mathcal{R}}(h, w)), & p_i \leq T_{cf} \\ p_i * \text{Avg}(X_j(h, w), ..X_{j+\mathcal{R}}(h, w)), & p_i > T_{cf} \end{cases} \tag{2}$$

where, $p_i$ is the $i^{th}$ element of $\mathbf{p}$, and $T_{cf}$ is channel fusion threshold. This is a hyperparameter which is set to 0.5 based on our ablation study (See Sec. 4). $Y_i(h, w)$ is the value at the $(h, w)$ spatial location of the $i^{th}$ channel in the output feature map $\mathbf{Y}$, $i \in [0, C/\mathcal{R}]$ and $j = \mathcal{R} * i$. A pictorial represntation of this fusion strategy is provided in Figure 3a.

Figure 3: **Channel Fusion strategies.** (a) Threshold based fusion, and (b) weighted fusion.

### 3.3.2 Weighted Fusion

In contrast to the threshold based fusion, this strategy is free of $T_{cf}$. In this strategy, for any spatial location $(h, w) \in H \times W$, all the $\mathcal{R}$ scalar values are first reduced by using both of the reduction operators separately, and then the resulting values are combined using a weighted sum as given below.

$$Y_i(h, w) = p_i * \text{Max}(X_j(h, w), ..X_{j+k}(h, w)) + (1 - p_i) * \text{Avg}(X_j(h, w), ..X_{j+k}(h, w)) \quad (3)$$

In this strategy, the network itself learns to determine how much contribution should be given to a reduction operator. Due to this flexibility, each location in a chunk can enjoy different combinations of the reduction operators. This is not the case with threshold based fusion. Infact, threshold based fusion becomes a special case of weighted fusion when $p_i$ approaches the two extremas i.e. $\{0, 1\}$. Figure 3b illustrates the weighted fusion strategy.

In both of the above techniques, the use of **p** helps choosing or weighting different reduction operators adaptively based on the input. Without that, all the squeeze layers shall have to be eliminated which will reduce to the case of (Hussain & Hesheng, 2019), leading to a heavy accuracy loss (Table 2).

### 3.4 A Theoretical Viewpoint

Theoretically, FACS does not introduce any parameter overhead instead reuse the parameters of the squeeze layer. As squeeze layer has learnable parameters, therefore it also contributes in the non-linearity of the network. Hence, FACS deals with two problems. *First*, preserve non-linearity, and *Second*, reduce computations. The computation reduction primarily occurs due to collapsing the input tensor $\mathbf{X} \in \mathbb{R}^{C \times H \times W}$ into a 1D descriptor $\mathbf{z} \in \mathbb{R}^C$. Global pooling can potentially introduce information loss, therefore loss of non-linearity. However, the proposed learnable cross channel information blending **p** and the fusion operator works in such a way that non-linearity is maintained.

In this process, role of **p** is crucial which is used for aggregating knowledge across a chunk of $\mathcal{R}$ channels at a given spatial location in a feature map to dynamically decide the choice of fusion operator. In case of max fusion, only one channel out of $\mathcal{R}$ goes out while in the AVG pooling, the output have an effect of all the channels in the chunk. Interestingly, each chunk of $R$ channels can dynamically undergo either Max or Avg fusion, depending on **p**. Hence, each spatial location has flexibility to choose among Max or AVG. For this reason, at any given spatial location $\in \mathbb{R}^{H \times W}$, any one of the $\mathcal{R}$ channels make it to the output in max pooling, while at another location, some other channel wins. This degree of freedom introduces a high level of non-linearity into the network which helps achieving FACS slightly better accuracy even at significantly lower FLOP requirements. These improvements brings huge training and inference speed-ups and are empirically verified in the Sec. 4. Effect of FACS on FLOPs and memory is discussed in appendix 5.

### 3.5 FACS instantiation

**FACS-ResNet.** ResNet is enhanced with FACS by replacing all the residual squeeze-expand units (Figure 4a) with FACS ones (Figure 4b). We experiment with ResNet-50, ResNet-101, and ResNet-152, and add "FACS" prefix to the resulting networks.

**FACS-VGG.** Although VGG (Simonyan & Zisserman, 2014) does not have a provision of squeeze layers since it consists only of $3 \times 3$ convolution (Figure 4c), we also pave a way to enhance them with FACS, due to its popularity in robotics and other real-time applications. To achieve that, we insert a $1 \times 1$ convolution before each $3 \times 3$ convolution layer except the first (Figure 4d). The purpose of this layer is to function as a squeeze layer which reduces the input channels by a factor of $\mathcal{R}$. We name the resulting model as VGG-$1 \times 1$ which is enhanced with FACS as shown in Figure 4e.

**FACS-MobileNet-v2.** We also study FACS for mobile networks MobileNet-v2 (Sandler et al., 2018) which employs inverted residual units with depthwise separable convolutions. The baseline inverted residual unit and its FACS enhanced counterpart is illustrated in Figure 4f, 4g.

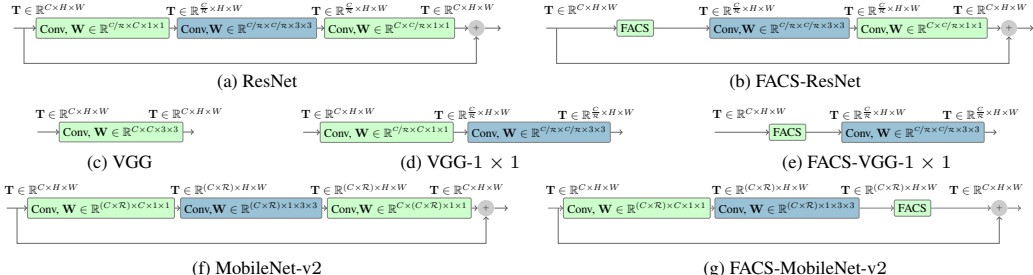

Figure 4: **Zoom in.** (a) Squeeze expand residual unit, (b) FACS variant, (c) VGG block, (d) VGG-1×1 block, (e) FACS variant, (f) inverted squeeze-expand unit, and (b) FACS variant.

## 4 EXPERIMENTS

We evaluate FACS on ILSVRC'2012 ImageNet (Deng et al., 2009) having 1.28M training and 50K validation images over 1000 categories. For illustration of transfer learning, we use CIFAR-10 and CIFAR-100 image classification datasets. We also evaluate FACS onto a downstream task of semantic segmentation against a popular pipeline (Zhao et al., 2017). We first show a comprehensive ablation study of FACS design, and then examine impact of FACS into previously instantiated networks. We also provide crucial throughput and latency improvements over a five different GPU hardware including edge devices. All the baselines are retrained in PyTorch (Paszke et al., 2019).

**Training Specifications.** The training procedure is kept straightforward to ensure reproducibility. We use a batch size of 256 which is split across 8 GPUs. We use a RandomResized crop of $224 \times 224$ along with horizontal flip. We use `SGD` with `Nesterov` momentum of 0.9, `base_lr`=0.1 with CosineAnnealing (Loshchilov & Hutter, 2016) learning rate scheduler, and a weight decay of 0.0001. All models are trained for 120 epochs following ResNet (He et al., 2016), unless otherwise stated.

### 4.1 ABLATION STUDY

We empirically validate FACS design practices by performing the most pertinent ablations possible. ResNet-50 is adopted as the baseline for this purpose. To begin with, we first analyze the effect of changing the activation function in the cross-channel information blending stage, and then examine the effect of placing a BatchNorm prior to the sigmoidal activation. Further, we verify the behavior of proposed channel fusion strategies, and also the effect of varying fusion threshold $T_{cf}$.

**Fusion Activation.** The channel fusion stage utilizes the fusion probability $\mathbf{p}$ produced by the sigmoidal layer. Given that the value of $\mathbf{p}$ lies in the interval $[0, 1]$, we wished to examine the behavior of FACS if this range is achieved via a different activation function. For this purpose, we select `TanH` function which natively squeezes the input into a range $[-1, 1]$. Therefore, we rewrite the mathematical expression to $0.5 * (1+\text{TanH})$ in order to place the output of TanH into the desired range of $[0, 1]$. We replace the sigmoidal activation with the above expression and retrain the network. The corresponding results are shown in the top section of Table 1. The measures reveal that sigmoidal activation is superior to TanH activation for the case of FACS.

**BatchNorm in Global Context Aggregation.** Out of curiosity, we also analyze the behavior of FACS module by placing a BatchNorm (Ioffe & Szegedy, 2015) after the $1 \times 1$ convolution, because the squeeze layer in the baseline method is also followed by a BatchNorm layer. We observe that BatchNorm negatively impacts the performance (second section, Table 1).

**Effect of Fusion Threshold ($T_{cf}$).** The hyperparameter $T_{cf}$ is evaluated against three values $\in \{0.0, 0.5, 1.0\}$. In accordance with the Eq. 2, $T_{cf} = 0$ corresponds to Avg operator, $T_{cf} = 1.0$ corresponds to Max operator regardless of the value of $\mathbf{p}$. Whereas $T_{cf} = 0.5$ offers equal opportunity to the Max and Avg reduction operators which is adaptively taken care of by the value of $\mathbf{p}$. We present an ablation over the aforementioned three values of $T_{cf}$. The third section of Table 1 holds the corresponding results. It is observable that $T_{cf} = 0.5$ attains the best performance, which is the case when the network has the flexibility to choose between the two reduction operators adaptively. Hence, in the subsequent experiments, we use $T_{cf} = 0.5$ for threshold based fusion. One may argue that there is marginal accuracy difference for different $T_{cf}$ values, so why not use a prefixed operator? As stated earlier in Sec 3.3, in that case $\mathbf{p}$ becomes of no use, and corresponding conv layer will have to be removed, which translates to (Hussain & Hesheng, 2019), resulting in loss of parameters and accuracy (Table-2).

Table 1: Ablation study of FACS-ResNet-50. Top-1 Accuracy on ImageNet.

| Ablation | Parameter | Top-1 Accuracy |
|---|---|---|
| Fusion Activation | Sigmoid | 76.77% |
| | TanH | 76.39% |
| Batch-Norm | ✗ | 76.77% |
| | ✓ | 76.44% |
| $T_{cf}$ | 0.0 | 76.58% |
| | 0.5 | 76.77% |
| | 1.0 | 76.54% |
| Channel Fusion Strategy | Threshold based Fusion (Eq. 2) | 76.77% |
| | Probabilistic Weighted Fusion (Eq. 3) | 76.52% |

**Channel Fusion Strategy.** We train the network for both of the fusion strategies. Our findings suggest that threshold based fusion is superior to the weighted fusion, as shown in the bottom section of Table 1. Although, the effect is marginal, in the context of implementation, Eq. (2) is easier and faster as compared to Eq. (3). It is because, Eq. (2) requires atmost one operator to be computed based on the value of $T_{cf}$, whereas Eq. (3) requires both of the operators to be computed.

## 4.2 IMAGENET CLASSIFICATION

To the best of our knowledge, computationally efficient squeeze operation has not been discussed in the literature, hence due to the lack of existing algorithms on this problem, the most suitable evaluation possible is with the baselines only. Nonetheless, we could find (Hussain & Hesheng, 2019) which is only the aligned work. We have compared our approach with this approach.

### 4.2.1 FACS-RESNET

Table 2 shows evaluation of FACS when plugged into ResNet. FACS indeed achieves computationally efficient squeezing, as indicated by the $\sim 23\%$ reduction in FLOPs of all the FACS variants. Though improving the accuracy is not our goal, FACS also boosts the accuracy significantly i.e. $0.47\%$, $0.45\%$, $0.44\%$ for FACS-50, 101, and 152 respectively. To our surprise, FACS-ResNet-101 surpasses the baseline ResNet-152 by $0.18\%$ in accuracy and $47\%$ in FLOPs, indicating a huge improvement.

**FACS-ResNet vs Depthwise/Channel Pooling.** In terms of comparison, a closely matching approach is depthwise pooling (DWP) (Hussain & Hesheng, 2019) which is a parameterless method. We endow ResNet-50 with DWP and retrain the network with identical settings of FACS-ResNet-50. Table 2 shows the corresponding comparison. It can be noticed that DWP drastically reduces the number of parameters of the baseline ResNet which results in a huge accuracy drop of $1.68\%$. Although, there is significant parameter gap between FACS/ResNet and DWP, therefore, at a first glance the comparison might seem unfair. However, our intention to conduct this experiment is to demonstrate that FACS does not incur any reduction in parameter while maintains the accuracy.

**Reduction Ratio ($\mathcal{R}$).** It is important to note that the reduction ratio $\mathcal{R}$ is not a hyperparameter of FACS. It can be a potential source of confusion when discussing (Hu et al., 2018; Woo et al., 2018) which employ two reduction ratios: one for the squeeze layer ($\mathcal{R}$) and one for the SE or CBAM modules ($r$). The value of $\mathcal{R}$ is independent of the FACS, SE or CBAM since it is the hyperparameter of the baseline ResNet where it is set to 4. Likewise, $\mathcal{R}$ is also set to 4 in the FACS variants. On the other hand, the reduction ratio $r$ is set to 16 in the SE unit of (Hu et al., 2018) and the shared MLP of the (Woo et al., 2018), but is irrelevant to FACS.

Although ablation on $\mathcal{R}$ is out of scope of this paper, yet we examine the impact of varying $\mathcal{R}$ on the performance of ResNet and FACS variants. To evaluate, we perform an additional experiment by setting $\mathcal{R} = 8$ whose outcomes are displayed in Table 2. It can be noticed that with $\mathcal{R} = 8$, FACS-ResNet performs better by $\sim 0.81\%$ with almost $25\%$ lesser FLOPs which is huge in itself. In addition, the accuracy gap between ResNet for $\mathcal{R} = 4$ and $\mathcal{R} = 8$ is $2.64\%$, while this gap reduces to $2.30\%$ for FACS at significant $56\%$ reduction in the computation cost. It shows the robustness of FACS towards parameter reduction and improved generalization over large scale ImageNet dataset by learning richer data representations. It further signifies the vast scope of FACS in edge devices, especially during network customization for a required purpose.

**FACS in Conjunction With SE.** When used in conjunction with SE (Hu et al., 2018) in ResNet-50, FACS reduces the computational demands of baseline SE-ResNet-50 while performing better, Table 2.

Table 2: FACS vs ResNet. FACS is better for a negative '$\Delta$Acc%' and a positive '$\Delta$#FLOPS%'.

| Architecture | $\mathcal{R}$ | #Params | #FLOPs | Top-1 | Top-5 | $\Delta$Acc% | $\Delta$#FLOPs% |
|---|---|---|---|---|---|---|---|
| DWP-ResNet-50 | 4 | 19.6M | 2.82B | 74.62% | 92.10% | – | – |
| ResNet-50 (He et al., 2016) | 4 | 25.5M | 4.12B | 76.30% | 92.69% | – | – |
| FACS-ResNet-50 | 4 | 25.5M | 3.18B | 76.77% | 93.34% | $-0.47\%$ | 22.8% |
| ResNet-101 (He et al., 2016) | 4 | 44.5M | 7.85B | 77.21% | 93.61% | – | – |
| FACS-ResNet-101 | 4 | 44.5M | 6.05B | 77.96% | 93.91% | $-0.45\%$ | 22.9% |
| ResNet-152 (He et al., 2016) | 4 | 60.1M | 11.58B | 77.78% | 93.80% | – | – |
| FACS-ResNet-152 | 4 | 60.1M | 8.91B | 78.12% | 94.02% | $-0.44\%$ | 23.0% |
| ResNet-50 | 8 | 12.3M | 1.85B | 73.66% | 91.59% | – | – |
| FACS-ResNet-50 | 8 | 12.3M | 1.39B | 74.47% | 92.07% | $-0.81\%$ | 24.8% |
| SE-ResNet-50 (Hu et al., 2018) | 4 | 28.0M | 4.13B | 76.71% | 93.38% | – | – |
| SE-FACS-ResNet-50 | 4 | 28.0M | 3.19B | 76.95% | 93.40% | $-0.24\%$ | 22.8% |

Table 3: Latency analysis of FACS vs ResNet @224 $\times$ 224, FP32 precision, averaged over 25 runs.

| Compute Platform | Cores | Compute power | Res-50 | FACS-50 | Res-101 | FACS-101 | Res-152 | FACS-152 |
|---|---|---|---|---|---|---|---|---|
| NVIDIA A40 | 10752 | 37.00 TFLOPs | 7ms | 6ms | 11ms | 10ms | 15ms | 14ms |
| NVIDIA RTX2080Ti | 4352 | 13.45 TFLOPs | 8ms | 6ms | 14ms | 12ms | 17ms | 15ms |
| NVIDIA GTX1080Ti | 3584 | 11.45 TFLOPs | 9ms | 7ms | 13ms | 12ms | 17ms | 15ms |
| NVIDIA Jetson NX | 384 | 1.000 TFLOPs | 48ms | 40ms | 75ms | 64ms | 100ms | 85ms |
| NVIDIA Jetson Nano | 128 | 0.235 TFLOPs | 140ms | 130ms | 230ms | 200ms | 320ms | 280ms |

**Throughput & Latency.** The $\sim 23\%$ FLOPs reduction directly reflects increased throughput as training time reduces from 4 to 2.5 days for ResNet-50, 4 to 3 days for ResNet-101, and 6 to 4 days for ResNet-152 on an $8\times$ NVIDIA 1080Ti system. Additionally, huge latency improvements are also seen on edge computing devices. Table 3 shows the latency analysis on a number of GPUs with varying compute capabilities. It is done in order to demonstrate the portability of FACS across devices with varying computational power. The first three are high power GPUs among which the first is compute specialized, second and third are gaming GPUs with former powerful than the latter. On the other hand, the last two are edge computing devices which are far less powerful.

The difference in latency is mostly attributed to the variation in the number of computing elements or cores. Theoretically, more cores should run a network faster, however owing to the sequential linking of layers, a layer must wait until preceding ones finish. This causes similar latency for the first three GPUs. However, for such devices, considerable gains are seen during batched inference and training, which is a measure of throughput and, is reflected via days-long reduction in training time.

In contrast, the impact of FACS is more pronounced on edge devices, where the cores are a scarce resource. On Jetson-NX, FACS-50 is 16% faster then ResNet-50, FACS-101 is 14% faster, and FACS-152 is 15% faster. On Jetson-Nano, FACS-50 is 7% faster then ResNet-50, FACS-101 is 13% faster, and FACS-152 is 12% faster. Notably, these speed-ups are huge which can be further enhanced via half-precision (FP16) inference, roughly doubling the speed. Considering the extensive usage of edge devices in real-time applications, the aforementioned enhancements are quite advantageous.

**Dynamic Pruning (Gao et al., 2018) vs FACS** Although dynamic pruning is not applicable to channel squeezing, we customize a recent approach "FBS" (Gao et al., 2018) for this purpose. FBS works by subsampling channels from the input via global pooling followed by a convolution to compute saliency, and then keeping top-k channels based on that, while turning off the remaining ones by setting all the values to zero. Notice that, in FBS, both input and output tensors lie in $\mathbb{R}^{C \times H \times W}$. To adapt it for our use, we keep the top-k channels, where k equals to $C/\mathcal{R}$. In this case, FBS outputs a tensor $\in \mathbb{R}^{\frac{C}{\mathcal{R}} \times H \times W}$ instead of $\in \mathbb{R}^{C \times H \times W}$ which is its default behavior.

During training, we see convergence issues. This happens because when keeping only top-k channels, the intermediate channels are dropped based on the input which can be different for different input. When this tensor is passed to the subsequent convolution, due to dynamically changing position of the channels in the output, the convolution struggles to learn the relation between the channels, because channel indexing is lost during choosing top-k. This problem does not occur in naive FBS. Secondly, FBS requires an additional convolution to compute channel saliency which has more parameters in contrast to of FACS. It leads to roughly 48% rise in the parameters i.e. from 24.4M to 37M, which is not desirable. This clearly indicates that pruning methods can not complement FACS, which further signifies novelty and utility of FACS.

Table 4: FACS-VGG vs VGG on ImageNet.

| Architecture | $\mathcal{R}$ | #Params | #FLOPs | Top-1 | Top-5 | $\triangle$Acc | $\triangle$#FLOPs |
|---|---|---|---|---|---|---|---|
| VGG-16-1 $\times$ 1 | 2 | 10.3M | 10.0B | 74.29% | 91.86% | — | — |
| FACS-VGG-16 | 2 | 10.3M | 9.3B | 74.82% | 92.16% | $-0.53\%$ | 7.0% |
| VGG-16-1 $\times$ 1 | 4 | 7.4M | 6.7B | 72.59% | 90.88% | — | — |
| FACS-VGG-16 | 4 | 7.4M | 6.3B | 72.71% | 91.01% | $-0.12\%$ | 6.0% |

Table 5: FACS for MobileNet-v2 vs MobileNet-v2 evaluation on ImageNet.

| Architecture | BatchNorm (Ioffe & Szegedy, 2015) | #Params | #FLOPs | Top-1 | Top-5 | $\triangle$Acc | $\triangle$#FLOPs |
|---|---|---|---|---|---|---|---|
| MobileNet-v2 (Sandler et al., 2018) | — | 3.50M | 0.32B | 71.80% | 90.19% | — | — |
| FACS-MobileNet-v2 | ✗ | 3.50M | 0.25B | 71.13% | 89.85% | 0.67% | 21.8% |
| FACS-MobileNet-v2 | Pre | 3.50M | 0.25B | 71.00% | 89.75% | 0.80% | 21.8% |
| FACS-MobileNet-v2 | Post | 3.50M | 0.25B | 70.79% | 89.59% | 1.01% | 21.8% |
| MobileNet-v2 (Sandler et al., 2018) | — | 25.2M | 4.65B | 74.48% | 92.09% | — | — |
| FACS-MobileNet-v2 | ✗ | 25.2M | 4.58B | 74.57% | 92.07% | $-0.09\%$ | 1.2% |
| FACS-MobileNet-v2 | Pre | 25.2M | 4.58B | 74.65% | 92.11% | $-0.17\%$ | 1.2% |
| FACS-MobileNet-v2 | Post | 25.2M | 4.58B | 74.52% | 92.17% | $-0.04\%$ | 1.2% |

### 4.2.2 FACS-VGG

We select the popular VGG-16 variant of VGG series and upgrade it to VGG-16-1 $\times$ 1 model with BatchNorm (Ioffe & Szegedy, 2015) in all the layers. We compare this model with its FACS variant. The experiments are conducted for two values of reduction ratio $\mathcal{R}$ i.e. $\{2, 4\}$, the results for which are shown in Table 4.

It can be observed that the FACS is consistent as computation reduction is $\sim 6\%$ while accuracy is improved significantly for different values of $\mathcal{R}$. Noticeably, relative gain w.r.t. ResNet is smaller ($23\%$ vs $6\%$), it is so because in this architecture most computations are packed into $3 \times 3$ convolutions ($90\%$) instead of $1 \times 1$ ($6\%$). Nonetheless, potential of FACS is huge in VGG like architecture, especially for low power applications and edge devices because narrower models of VGG are quite popular for real-time applications (Kumar et al., 2020).

### 4.2.3 FACS-MOBILENET-V2.

Mobile networks are not our point of consideration because of their limited representation power and complexity, as mentioned in Sec. 2. However, we analyze how FACS behaves with mobilenets which employ inverted-residual bottlenecks consisting of depthwise separable convolution networks. We choose MobileNet-v2 (Sandler et al., 2018) for the evaluation purpose and train for 120 epochs.

The experimental results are reported in Table 5. We observe that for depthwise separable convolution, FACS performs slightly inferior. We hypothesis that the accuracy drop is due to the lack of inter-channel communication in depth-wise separable convolutions. In MobileNet-v2, this communication is performed by the squeeze layer in addition to performing the task of reducing the number of channels. However in FACS, this is performed by global pooling and the convolution layer which operates on a descriptor (less information) instead of the tensor (more information) which according to our hypothesis becomes the reason of slightly reduced accuracy.

To validate our hypothesis, we replace all the depthwise separable convolution by fully connected convolutions in MobileNet-v2. In this case, we observe improvements in the accuracy, as shown in the second section of Table 5. This verifies our previously made hypothesis that devoid of inter-channel communication limits the performance of FACS. However, in this case the FLOPs improvement drops which is expected. It is so because, inverted residual design increases the number of channels which when fed to a fully connected $3 \times 3$ convolution, leads to most of the network computations packed into these layers ($95\%$). Hence, effect of FACS diminishes. This experiment is for hypothesis verification only because inverted bottlenecks with fully connected $3 \times 3$ are not used in practice due to exponential rise in computations.

In addition to the above, we also performed ablations on MobileNet-v2 by placing BatchNorm (Ioffe & Szegedy, 2015) at the input and output of FACS. We observe that BatchNorm slightly lowers the performance of depthwise convolutions while for the fully connected case, pre-BatchNorm marginally improves the performance. Conclusively, we say that FACS performs on a par for depthwise separable convolutions, but $21\%$ reduction in FLOPs can be traded for an accuracy drop of $0.67\%$ which is negligible. While FACS performs better for fully connected convolutions in inverted-residual units.

Table 6: FACS vs ResNet on classification, and semantic segmentation transfer learning.

| Architecture | CIFAR-10 | CIFAR-100 | CityScapes |
|---|---|---|---|
| ResNet-50 (He et al., 2016) | 95.57% | 81.60% | – |
| FACS-ResNet-50 | 95.67% | 82.22% | – |
| PSPNet (Zhao et al., 2017) + ResNet-101 (He et al., 2016) | – | – | 78.4% |
| PSPNet (Zhao et al., 2017) + FACS-ResNet-101 | – | – | 79.1% |

Figure 5: GradCAM visualizations. More red shows more confidence for a pixel to belong to a class.

## 4.3 TRANSFER LEARNING

To analyze the generalization of FACS across datasets, we perform transfer learning from ImageNet to CIFAR-10 and CIFAR-100. Each of the dataset consists of 50K training and 10K test images. For training, we finetune FACS-ResNet-50 and ResNet-50 pretrained over ImageNet. The training protocol for both the datasets remain identical to that of ImageNet except 200 epochs. Table 6 shows the classification results. It can be seen that, surprisingly, FACS-ResNet-50 outperforms the baseline by $0.10\%$ on CIFAR-10 and by $0.62\%$ on CIFAR-100 at a lower FLOP requirements.

In addition, we evaluate FACS for a challenging downstream task of semantic segmentation. We use the popular approach (Zhao et al., 2017) and replace the backbone with FACS-ResNet-101. Consistently, FACS variant outperforms the baseline both in terms of FLOPs efficiency and accuracy by $0.7\%$ units mIoU (mean intersection over union), indicating that FACS indeed transfers better across tasks and datasets.

## 4.4 GRADCAM VISUALIZATION

As FACS employs global context in one of its stages and also choose the reduction operators adaptively, it encouraged us to analyze the behavior of layer activations in the network corresponding to a groundthruth class. To do this, we use GradCAM (Selvaraju et al., 2017) that utilizes gradients to visualize the importance of spatial activations. GradCAM is given a label for which the attended regions in the feature maps are computed. This feature of GradCAM is utilized to visualize and understand that how FACS attends the spatial regions in comparison to the baseline networks. It gives us the qualitative explanation that despite the reduction in the computations, why FACS is performing

Figure 5 shows the GradCAM analysis for ResNet and VGG. It can be seen that FACS significantly improves the attended regions of the target class relative to the baselines. Also, in case of images with multiple instances of an object, FACS focuses on each instance strongly. It indicates that FACS enhances a network's ability to generalize better by learning to emphasize class-specific parts.

## 5 CONCLUSION

In this work, we introduce a novel module called "Fast Adaptive Channel Squeezing" (FACS) which redefines the manner in which channel squeezing is accomplished in CNNs. It functions by exploiting global spatial context and blending cross-channel information in order to carry out computationally efficient channel squeezing. We plugged the FACS module into three distinct CNN architectures: ResNet, VGG, and MobileNet-v2, and presented a comprehensive ablation study including ImageNet classification results and transfer learning on semantic segmentation task. Our experimental study suggests that FACS-enhanced CNNs are roughly $23\%$ faster (lower FLOPs) while being more accurate in the same number of parameters while generalizing well across datasets and tasks. We show that the reduction in FLOPs cuts a network's training time significantly, and also demonstrate that FACS results in lower latency of a network which speeds-up inference on edge devices.

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

APPENDIX

# A  COMPUTATIONAL COMPLEXITY

The primary intent of the FACS module has been to reduce the computational complexity of squeeze operation. Therefore, it is necessary to discuss the computational requirement associated with FACS. However for better understanding, we first discuss the FLOPs of different kinds of layers.

## A.1  CONVOLUTION

Consider a convolution layer having $N$ kernels and an input feature map $\mathbf{X} \in \mathbb{R}^{C \times H \times W}$. The size of each kernel can be given by $C \times k \times k$. FLOPs for convolution operation are determined in terms of Fusion-Multi-Addition (FMA) instructions, therefore the computational demands of a convolution layer can be given as:

$$\#\text{FLOPs} = H \times W \times N \times C \times k \times k \tag{4}$$

## A.2  BATCHNORM

The BatchNorm Ioffe & Szegedy (2015) operation is performed per spatial location and can be given as $\hat{\mathbf{X}} = (\mathbf{X} - \mu)\frac{\gamma}{\sigma} + \beta$. It can be implemented in three FLOPs i.e. first for computing $X - \mu$, second for $\gamma/\sigma$ and last as FMA with $\beta$. In general, $\sigma$ is stored as $\sigma^2$, therefore, it requires to compute square-root of $\sigma^2$ to obtain $\sigma$. Overall, it takes four FLOPs to implement a BatchNorm operation per spatial location. Thus, total number of FLOPs for a BatchNorm layer can be given as:

$$\#\text{FLOPs} = 4 \times C \times H \times W \tag{5}$$

Optionally, during inference, BN can be fused with a Conv operation where a convolution is followed by BN, but we remain agnostic to such cases to account for the training phase and other architectures.

## A.3  RELU

A ReLU operation is given by $\mathbf{Y} = X$ for $X \geq 0$ and $\mathbf{Y} = 0$ for $X < 0$. It simply requires a comparison instruction, leading to total number of FLOPs given by:

$$\#\text{FLOPs} = C \times H \times W \tag{6}$$

## A.4  SIGMOID

A Sigmoid operation is given by $\mathbf{Y} = 1/1+\exp^{-\mathbf{x}}$. It can be implemented in four FLOPs. Therefore, the total FLOPs for a Sigmoid layer can be given by:

$$\#\text{FLOPs} = 4 \times C \times H \times W \tag{7}$$

## A.5  GLOBAL POOLING

Apart from the above layers, in an FACS module, a global pooling operation is also performed. There are several ways to implement a global pooling operation, however, the most common is by using matrix multiplication routines and FMA instructions. The whole channel of a feature map can be considered as a vector of size $H \times W$ which can be reduced to a scalar by taking its dot product with a vector whose all of the elements equals to one. Hence, the total number of FLOPs for the global pooling operation can be given by:

$$\#\text{FLOPs} = C \times H \times W \tag{8}$$

## A.6  CHANNEL FUSION

Channel fusion operates on a chunk of $k$ channels. For a Max operation, $(k-1)$ compare instructions, while for an Avg operation, $(k-1)$ FMA instructions are required. Thus, the total number of FLOPs for channel fusion can be given by:

$$\#\text{FLOPs} = (k-1) \times (C/k) \times H \times W \tag{9}$$

Based on the several equations developed above, the computational complexity of residual bottleneck and the FACS block can be calculated.

## A.7 Effect of FACS on Computational Complexity

In the baseline method, the squeeze layer operates upon $\mathbf{X} \in \mathbb{R}^{C \times H \times W}$ which requires $C/\mathcal{R} \times C \times H \times W$ FLOPs. Whereas in FACS, the global context aggregation requires $C \times H \times W$ FLOPs, cross-channel information blending requires $C/\mathcal{R} \times C$ FLOPs. and channel fusion requires $C/\mathcal{R} \times (\mathcal{R} - 1) \times H \times W$ FLOPs.

As an example, consider an input tensor $\mathbf{X} \in \mathbb{R}^{12 \times 5 \times 5}$ to a squeeze layer kernels of size $1 \times 1$. With $\mathcal{R} = 4$, the $N = 3$. From the equations below, total number of FLOPs for a squeeze layer equals to 8475.

$$\#\text{Conv\_FLOPs} = 5 \times 5 \times 3 \times 12 \times 1 \times 1 = 900 \tag{10}$$

$$\#\text{BN\_FLOPs} = 4 \times 3 \times 5 \times 5 = 300 \tag{11}$$

$$\#\text{ReLU\_FLOPs} = 3 \times 5 \times 5 = 75 \tag{12}$$

On the other hand, based on the following equations, the FLOPs for the FACS module with $k = 4$ equals to 811.

$$\#\text{Pooling\_FLOPs} = 12 \times 5 \times 5 = 250 \tag{13}$$

$$\#\text{Conv\_FLOPs} = 1 \times 1 \times 3 \times 12 \times 1 \times 1 = 36 \tag{14}$$

$$\#\text{Sigmoid\_FLOPs} = 4 \times 3 \times 1 \times 1 = 12 \tag{15}$$

$$\#\text{Fusion\_FLOPs} = 3 \times 3 \times 5 \times 5 = 225 \tag{16}$$

In the above example, the baseline squeezing method requires 1275 FLOPs, whereas FACS requires only 523 and 748 FLOPs for threshold based and weight fusion respectively. In the similar manner, we achieve huge gains when FACS is plugged into the existing networks which are discussed in the experiments. For the calculation of FLOPs, kindly see above.

## A.8 Effect of FACS on Memory

Despite the computational benefits, FACS does not introduce any memory overhead. The total memory required by the baseline squeeze operation can be given by: $\#\text{M} = C/4 \times H \times W$. On the other hand, the memory required for FACS is given by: $\#\text{M} = C + C/4 + C/4 \times H \times W$. We can see that there is negligible increment in the memory footprint i.e. from $0.75 \times C \times H \times W$ to $0.75 \times C \times H \times W + 1.25C$. For an FP32 precision, the raw memory footprint will be $4 \times M$.

## B Codes and Implementation

We will opensource the codes and all the pretrained models for the framework in PyTorch (Paszke et al., 2019). In order to better understand the concept, we have included our GPU implementation of channel fusion layer in the supplementary material. The corresponding file is named as `facs_layer.cu` which consists of GPU implementation of the fusion formulations described in the paper. Other files `facs_layer.cpp`, `facs_layer.py` are also provided which holds pytorch class implementation of FACS. We have not provided the network training and inference code which will be released later.

It also contains code for the new experiment related to FBS in three files: `facs_fbs_layer.cu` which consists of GPU implementationa along with helper files `facs_fbs_layer.cpp`, `facs_fbs_layer.py`

## C GPU Considerations

The implementation of FACS operator is quite straightforward and fully parallelizable. For reference, we also have provided the GPU implementation in the supplementary material. The computations of fusion probability and the fused feature map are entirely parallelizable because they are pointwise operations. The whole code can be implemented in merely $10 - 15$ lines of a CUDA kernel and other parallelization device.

