# OpenReview forum: "FACS: FAST ADAPTIVE CHANNEL SQUEEZING"
_ICLR.cc/2023/Conference — Submitted to ICLR 2023_

### Official Review · Reviewer_ZY5e · 2022-10-19

**Confidence:** 5
**Clarity, Quality, Novelty And Reproducibility:** 1. The writing is not good.

2. Lacks…
**Correctness:** 2
**Technical Novelty And Significance:** 1
**Empirical Novelty And Significance:** 1
**Recommendation:** 5

**Strength And Weaknesses:**

Weaknesses:

1. The reference style is problematic.

2. In Sec2.1, the comparison on number of training epoch dose not equal to the comparison of training efficiency, since the compared models are different.

3. Most of the 1x1 conv layers in ResNet are used for increasing the non-linearity while preserving the resolution of feature, it cannot be treated as channel squeezing.

4. The 'Global Context Aggregation' operation collapse the representation into single dimension, which must leads to spatial information loss.

5. In Sec 3.3.1, why the lower possibilities correspond to the average pooling while the higer ones correspond to max pooling? Any analysis and experiments for this operation?

6. In Sec3.3, the proposed combination manner is totally different from the existing one, since the combination of max and mean pooling should cover only a small region of the previous representation combination manner. However, there is no analysis and explanation to reason the proposed manner, which is hard to follow and understand.

**Summary Of The Paper:**

The authors propose to reduce the computation cost of deep learning model by reformulating the channel squeezing operation. Specifically, the 1x1 conv operation is replaced by firstly channel-wise average pooling the input feature, then calculated the called 'fusion possibilty'. Finally, the input channels are squeezed to be target size with the output 'fusion possibilty'.

**Summary Of The Review:**

The proposed method is far different from the exiting deep learning working scheme, and lacks of corresponding explanation and analysis, which is hard to believe the reported performance.

---

> ### Author Response · Authors · 2022-11-15
> **FACS is entirely novel approach which uses fusion in channel dimension adpatively**
>
> Dear reviewer, thank you for reading our paper. We reuse the 1x1 convolution in the FACS operation instead of incorporating newer layers which is the major uniqueness of FACS. Despite the significant experimentation, there has been a few concerns which we address below.
>
> **1. The reference style is problematic.**
>
> We have updated the reference style.
>
> **2. In Sec2.1, the comparison on number of training epoch dose not equal to the comparison of training efficiency, since the compared models are different.**
>
> e agree that the compared models are different, However, we wanted to make a point that in order to achieve comparable performances, the mobile series needs to train for very long schedules. And if the baseline or large networks trained for such schedules with appropriate data augmentation scheme, can easily surpass the smaller networks. Hence, representation power is of great importance for a given problem. If we can run a high presentation power network faster, we can achieve similar performance but in lower runtime as demonstrated in our experiments with 25\% reduction in FLOPs. It also improves training time drastically and latency on 5 different hardware platforms.
>
> **3. Most of the 1x1 conv layers in ResNet are used for increasing the non-linearity while preserving the resolution of feature, it cannot be treated as channel squeezing.**
>
> The 1x1 convolution essentially contains parameters, therefore it is indeed true that they add non-linearity to the network. However, their primary utility for which they were employed was to reduce the number of channels (channel squeezing) before feeding the 3x3 layers, as mentioned in the ResNet paper. We definitely keep account of the non-linearity and this is why our FACS is novel that it maintains the non-linearity of the overall network which is evident via improved accuracy  but at dramatically lower FLOPs i.e. 25%. In this sense, our FACS targets this problem of channel squeezing in a novel way and achieves state-of-the-art results.
>
>
> **4. The 'Global Context Aggregation' operation collapse the representation into single dimension, which must leads to spatial information loss.**
>
> We agree that it may lead to spatial information loss, but it depends on the way the global information is used, For example, it is used at the end of the classification network at the final layer to aggregate spatial knowledge (mentioned in Sec 3.1). In that case it improves the accuracy and prevents fully connected layers and overly large parameters at the final stage by avoiding the use of flattening which was used in VGG, increasing their parameters drastically, even more than the backbone in just two layers. It shows the success of global context aggregation.
>
> Further, post global context, we use a 1x1 convolution which introduces non-linearity into the network. Then this information is used to fuse channels from the input tensor using dynamically chosen fusion operators which are dynamically decided based on the fusion probability. This further adds non-linearity into the network. Together the channel fusion and the global context extraction and cross channel information blending helps maintaining the non-linearity of the network similar to the baseline, even at 25% lower FLOPs but slightly improved accuracy. Hence we say that in our case, global context extraction helps lowering the computational complexity, while non-linearity is achieved by the combination of several components of FACS altogether.
>
> **5. In Sec 3.3.1, why the lower possibilities correspond to the average pooling while the higer ones correspond to max pooling? Any analysis and experiments for this operation?**
>
> It is just a matter of choice that we implement to choose the fusion operators this way. Even if the order is reversed it does not matter, because a fusion operator is chosen based on the threshold for which ablation is provided, best at 0.5. Hence, if the network wishes to choose Max operator, it can choose it by increasing the p above the threshold, and vice-versa. Now if the order is reversed, this time network can reduce p to choose Max fusion operator. Hence, it is just an implementation choice which does not affect the performance.
>
> We have already performed this experiment, as it is just a programming or coding practice, it does not make a difference.

---

> > ### Author Response · Authors · 2022-11-15
> > **Contd..**
> >
> > **6.  In Sec3.3, the proposed combination manner is totally different from the existing one, since the combination of max and mean pooling should cover only a small region of the previous representation combination manner. However, there is no analysis and explanation to reason the proposed manner, which is hard to follow and understand.**
> >
> > Indeed our channel fusion in entirely novel. We believe that the potential source of confusion comes from the fact that, typically pooling operations are used in spatial dimension i.e. height and width dimension. Whereas in FACS, this occurs in channel dimension. The fusion operation essentially is performed at each spatial location in height and width dimension for each chunk of k-channels. This is entirely novel method of fusion, therefore may be difficult to understand in the beginning, however it is straightforward. We have tried our best to explain the same in Sec~3.3 *In this stage, the task of channel squeezing of X is performed in an adaptive and computationally efficient manner. We define two reduction operations: maximum (Max), and average (Avg). During fusion, the input feature map is first virtually partitioned into chunks of R channels, and then the channels of each chunk are adaptively fused to produce a single channel. In other words, fusion operation results in C/R output channels, if the number of input channels were C. We devise two strategies to perform this task, as discussed below.*
> >
> > We apologies that it was difficult to understand the FACS operation. Hence, now we have added slightly better explanation in the paper which makes it easier to understand. For ease, we write the explanation below.
> >
> > The first and foremost step of FACS is to gather global spatial context which is inspired by the fact that it is most frequently employed at the end of the classification networks to aggregate the knowledge of all spatial locations of the final feature map into a 1D descriptor. It is also utilized by to compute spatial statistics. Global pooling essentially can work as aggregating a whole channel and it is operated upon by a convolution layer with sigmoidal activations. The sigmoidal activations essentially are used for aggregating knowledge across chunk of k-channels at a given spatial location in a feature map. Then based on the sigmoidal activation, either max or avg aggregation is done using threshold or weighted fusion technique.
> >
> > In the overall operation, the dominant channels in a particular chunk are forwarded to the next step. In max pooling only one of the channel output goes out while in the AVG pooling, all the channels in the chunk are summed. Interestingly, each spatial location has flexibility to choose among different aggregation operation i.e. Max or AVG. Hence, at one location, any one of the k-channels can make it to the output in max pooling while at another location, some other channel makes it to the output. This flexibility or degree of freedom to choose dynamically add a high level of non-linearity into the network which helps achieving FACS slightly better accuracy even at 25\% lower FLOP requirements. These improvements brings huge training and inference speed-ups as shown in the experiment section-4.2.1 for different hardwares.
> >
> >
> > **The writing is not good.**
> >
> > We have improved the writing significantly. However, we would like to know from your expertise that what are the specific sections which you found inappropriate to read.
> >
> > **Lacks of analysis and explanation.**
> >
> > We have added the theoretical explanations. Although it was there, we wrote it along with each component. Now we have added and extra paragraph which only deals which theoretical aspect of FACS.

---

> > > ### Author Response · Authors · 2022-11-15
> > > **Contd...**
> > >
> > >
> > > **Limited novelty.**
> > >
> > > We sincerely say that the problem statement which FACS targets is both new and of practical importance, and has not been addressed previously. FACS is based on very elementary blocks of neural networks such as global pooling, convolutions which are also used in large spectrum of networks. However there combination is novel in FACS for the novel problem, as mentioned in Sec. 3. The channel fusion strategy in an adpative manner is novel which helps achieving faster execution while maintaining accuracy. Moreover, there are a few recent approaches which are orthogonal to FACS but still are based only upon simple concepts of global pooling and convolutions such as Dynamic Channel Pruning: Feature Boosting and Suppression, ICLR, 2019, but achieves good results in the area of pruning. Similarly FACS also achieves significant improvements which are of practical importance. Hence, we say that, novelty issue of FACS is not justified.
> > >
> > > Our FACS presented in the paper is described with ample figures and mathematics and achieves significant results. We also have reported analysis for different hardwares i.e. 5 different GPU platforms including heavy duty desktop grade to low power edge computing devices which itself quite challenging to report. We also have provided insight to internal representations of learnt by FACS and shown that they are much better. Being a very new concept for new problem, we respectfully say that question of novelty for FACS is not justified. In all we demonstrate state-of-art results on the novel problem, in a novel way, with significant empirical evidences.
> > >
> > >
> > > **The proposed method is far different from the exiting deep learning working scheme, and lacks of corresponding explanation and analysis, which is hard to believe the reported performance.**
> > >
> > > We respectfully say that it appears so because we targetted a new problem which is not addressed before. Hence, it may seem uncommon. However, if looked closely, our work closely is based on the fundamental concepts of neural networks which works extremely well. In addition, we have supported FACS with extensive experiments, every ablation possible, which clearly justifies the importance, significance of FACS and its utility.
> > >
> > > In addition, we also have shared a few codes and are open to release the pretrained models and the whole codes, which directly addresses your concern of  hard to believe the achieved results.
> > >
> > >
> > > Dear reviewer, Hopefully we have answered all of your questions. Please feel free to write back as your suggestions are extremely important for us to improve the quality of this paper.

---

> > > > ### Author Response · Authors · 2022-11-17
> > > > **New Draft of the paper uploaded**
> > > >
> > > > Dear reviewer,
> > > >
> > > > We have updated the paper. Due to change in citation style, we had to make adjustments to the text.  The changes are as given below.
> > > >
> > > > 1. Introduction slightly modified. Now mentioning that channel squeezing is important.
> > > > 2. Related work updated. Now contains discussion on dynamic pruning.
> > > > 3. A theoretical viewpoint added in Sec. 3.4 (new draft). To accommodate, the effect on memory and computational complexity example has been moved to appendix. Also the FLOP equations from the supplementary material has been moved to appendix in the paper for ease of reference.
> > > > 4. "Potential similarity.." Sec 3.6 in the initial submission has been moved in related work section to save space.
> > > > 5. Figure-4 updated with detailed dimensions of the tensors and colors.
> > > > 6. A discussion on the new experiment on dynamic pruning has been added to the experiments.
> > > > 7. A few spelling mistakes also have been corrected.
> > > > 8. Supplementary material now contains the code for FACS and also FBS for squeezing.
> > > >
> > > > Kindly go through the paper, and please let us know that it addressed all of your concerns.

---

> > ### Comment · Reviewer_ZY5e · 2022-11-25
> > **Discussion**
> >
> > 1. For response 2: Is there any comparision between the (computation complexity X iterations)? Otherwise the comparision is still not fair and hard to prove the efficiency of the proposed method.
> >
> > 2. I have updated my score from 3 to 5. However, I am still unsure about the novelty of the paper.

---

### Official Review · Reviewer_Ywmp · 2022-10-23

**Confidence:** 4
**Correctness:** 3
**Technical Novelty And Significance:** 2
**Empirical Novelty And Significance:** 2
**Recommendation:** 5

**Clarity, Quality, Novelty And Reproducibility:**

The reference style is wrong. The write-up is not very clear.

The novelty is limited

The authors provided the code in supplemental materials

**Strength And Weaknesses:**

Pros:

The proposed FACS method is able to reduce the computation cost and improve performance.

Cons:

(1) In my opinion, the global pooled feature will have the same effect on the input feature (with respect to performance), compared with the baseline squeezing. So the performance improvement should be caused by the channel fusion strategies. The authors are suggested to add one ablation study to show the effect of channel fusion strategies (e.g., a single type of pooling).

(2) The operations in the FACS block are very similar to that in SEnet (1 x 1 Conv on c x 1 x 1 feature is the same with the Fc layer).

(3) For Sec 3.3.1 and Sec 3.3.2, why output channel from the channel squeezing can be aligned with the channel-wise avg/max pooling?

(4) Lack of theoretical analysis of why the FACS and channel fusion module are useful.




**Summary Of The Paper:**

This paper proposes a FACS network to reduce the computation cost of deep learning models by redefining the module in channel squeezing. The original channel squeezing module (1x1 convolution) is amended by adding the global pooling and designing the channel fusion strategies.

**Summary Of The Review:**

As mentioned in 'Strength And Weaknesses', the proposed method novelty is not sufficient. Additionally, it also lacks of theoretical analysis. So I tend to give the 'marginally below the acceptance threshold'.

---

> ### Author Response · Authors · 2022-11-15
> **FACS is novel approach for the novel problem**
>
> Dear reviewer, thank you for the reviewing our paper. Despite the strengths, you have a few concerns which we try our best to resolve here.
>
> **1. In my opinion, the global pooled feature will have the same effect on the input feature (with respect to performance), compared with the baseline squeezing. So the performance improvement should be caused by the channel fusion strategies.**
>
> We respectfully say that this is not valid. The major computation saving occurs first from global pooling because in the baseline squeezing, a convolution operation is used which requires NxCxHxW computations where N is the number of kernels, C is the number of input channels and H,W heigh and width of a feature map. As stated in the supplementary material (in FLOP calculations) as an example, we revisit the example.
>
> Consider and input tensor 12x5x5 (CxHxW) to a squeeze layer kernels of size 3x3. With R=4, the
> N=3. From the FLOP equations mentioned in the supplementary material, total number of FLOPs for the baseline squeezing quals to 8475.
>
> Conv\_FLOPs = 5x5x3x12x3x3=8100
>
> BN\_FLOPs = 4x3x5x5=300
>
> ReLU\_FLOPs = 3x5x5=75
>
> Whereas for FACS, total FLOPs equals to 811.
>
> Pooling\_FLOPs = 12x5x5 = 250
>
> Conv\_FLOPs = 1x1x3x12x3x3 = 324
>
> Sigmoid\_FLOPs = 4x3x1x1 = 12
>
> Fusion\_FLOPs = 3x3x5x5 = 225
>
> From the above distribution, it can be seen that major computations saving occur from global pooling because now the convolution layer have alot less multiplications and additions. It can be seen that there is huge FLOP requirement difference between the squeeze layer and the FACS module.
>
>
> **The authors are suggested to add one ablation study to show the effect of channel fusion strategies (e.g., a single type of pooling).**
>
> Dear reviewer, we already have added an ablation of channel fusion strategy in Table-1 which shows that threshold fusion is slightly better and also faster as described in Sec. 4.1 under channel fusion strategy.
> Another ablation with threshold shows the effect of individual pooling. Tcf =0 means avg fusion, Tcf=1 means max fusion. The whole process is described in Sec4.1
>
> **2 The operations in the FACS block are very similar to that in SEnet (1 x 1 Conv on c x 1 x 1 feature is the same with the Fc layer).**
>
> Dear reviewer, we respectfully disagree because of the following major differences, as stated in Sec 3.6 , and SENet and CBAM are very different approaches for very different purpose:
>
> 1. The primary difference is the utility. FACS targets a novel problem of fast channel squeezing by reducing FLOPs requirements while maintaining the parameters and not sacrificing accuracy of the network. Whereas SENet and CBAM improves network accuracy by incorporating more parameters into the network and slightly increased FLOP requirement. This differentiate FACS from SENEt and CBAM at the first place.
>
> 2. Even structurally, FACS is extremely different from both SENet and CBAM. For example, SENet add two new convolution layers which are connected sequentially with ReLU and sigmoidal activation. One squeezes by a factor of 16 while another expands by the same factor and later input and the resulting vector are multiplied. On the other hand, FACS reuses the parameters of the already existing squeeze layer and does not add new layers.  Moreover, SENet or CBAM does not have a provision of fusion in channel dimension.
>
> 3. Most importantly, SE/ CBAM can function together as demonstrated in experiments 4.2.1.
>
> 4. Our intention to keep the paragraph was to strictly avoid confusions to the reader, since FACS uses common terms such as global pooling and squeezing. However, these approaches are orthogonal to FACS.
>
>
> **3 For Sec 3.3.1 and Sec 3.3.2, why output channel from the channel squeezing can be aligned with the channel-wise avg/max pooling?**
>
> We apologize but this question is not very much clear to us. However, if you mean that why the output of baseline channel fusion is aligned to the fusion strategy, we say that channel squeezing in baseline is a 1x1 convolution layer which along with reduces the number of channels, also aids in non-linearity of the overall network. Hence, if we use only channel-wise pooling, first the number of parameters reduces, as mentioned at the end of Sec. 3.3.2 that *In both of the above techniques, the use of p is inspired by the fact that it helps choosing or weighting different reduction operators adaptively based on the input. Without that, all the squeeze layers shall have to be eliminated which will reduce to the case of channel pooling, leading to a heavy accuracy loss (Table 2)*. This happens because the non-linearity introduced due to the squeezing layer gets eliminated by removal of the squeeze layers and Hence, drop in the performance.
>
> This is where FACS plays its role in a novel way where it performs the channel squeezing operation in computationally efficient manner, while keeping the parameters and the non-linearity which we empirically show by improved accuracies in Table-2.

---

> > ### Author Response · Authors · 2022-11-15
> > **Contd...**
> >
> >
> > **4 Lack of theoretical analysis of why the FACS and channel fusion module are useful.**
> >
> > We apologize that the text was not very much clear to you in this context. Instead of describing a unified theory, we have described the theoretical functioning and motivations of global context extraction, cross channel information blending and channel fusion in their respective sections. However, we now have improved the text and would like to share with you some theoretical aspects.
> >
> > The first and foremost step of FACS is to gather global spatial context which is inspired by the fact that it is most frequently employed at the end of the classification networks to aggregate the knowledge of all spatial locations of the final feature map into a 1D descriptor. It is also utilized by to compute spatial statistics. Global pooling essentially can work as aggregating a whole channel and it is operated upon by a convolution layer with sigmoidal activations. The sigmoidal activations essentially are used for aggregating knowledge across chunk of k-channels at a given spatial location in a feature map. Then based on the sigmoidal activation, either max or avg aggregation is done using threshold or weighted fusion technique.
> >
> > In the overall operation, the dominant channels in a particular chunk are forwarded to the next step. In max pooling only one of the channel goes out while in the AVG pooling, all the channels in the chunk are summed. Interestingly, each spatial location has flexibility to choose among different aggregation operation i.e. Max or AVG. Hence, at one location, any one of the k-channels can make it to the output in max pooling while at another location, some other channel makes it to the output. This flexibility or degree of freedom to choose dynamically add a high level of non-linearity into the network which helps achieving FACS slightly better accuracy even at 25% lower FLOP requirements. These improvements brings huge training and inference speed-ups as shown in the experiment section-4.2.1 for different hardwares.
> >
> > **The reference style is wrong. The write-up is not very clear.**
> >
> > We have updated the reference style. and improved the overall writeup of the paper. However it would be great if you can specifically pointout which sections to improve particularly
> >
> > **The novelty is limited**
> >
> > The problem statement which FACS targets is both new and of practical importance, and has not been addressed previously. FACS is based on very elementary blocks of neural networks such as global pooling, convolutions which are also used in large spectrum of networks. However there combination is novel in FACS for the novel problem, as mentioned in Sec. 3. Moreover, there are a few recent approaches which are orthogonal to FACS but still are based only upon simple concepts of global pooling and convolutions such as Dynamic Channel Pruning: Feature Boosting and Suppression, ICLR, 2019, but achieves good results in the area of pruning. Similarly FACS also achieves significant improvements which are of practical importance. Hence, we say that, novelty issue of FACS is not justified.
> >
> > Our FACS presented in the paper is described with ample figures and mathematics and achieves significant results. We also have reported analysis for different hardwares i.e. 5 different GPU platforms including heavy duty desktop grade to low power edge computing devices which itself quite challenging to report. We also have provided insight to internal representations of learnt by FACS and shown that they are much better. Being a very new concept for new problem, we respectfully say that question of novelty for FACS is not justified.
> >
> > **The authors provided the code in supplemental materials**
> >
> > Full codes along with the pretrained models shall be released post the review and respective deadlines of ICLR.
> >
> > Dear reviewer, Hopefully we have answered all of your questions. Please feel free to write back as your suggestions are extremely important for us to improve the quality of this paper.

---

> > > ### Author Response · Authors · 2022-11-17
> > > **New draft of the paper uploaded**
> > >
> > > Dear reviewer,
> > >
> > > We have updated the paper. Due to change in citation style, we had to make adjustments to the text.  The changes are as given below.
> > >
> > > 1. Introduction slightly modified. Now mentioning that channel squeezing is important.
> > > 2. Related work updated. Now contains discussion on dynamic pruning.
> > > 3. A theoretical viewpoint added in Sec. 3.4 (new draft). To accommodate, the effect on memory and computational complexity example has been moved to appendix. Also the FLOP equations from the supplementary material has been moved to appendix in the paper for ease of reference.
> > > 4. "Potential similarity.." Sec 3.6 in the initial submission has been moved in related work section to save space.
> > > 5. Figure-4 updated with detailed dimensions of the tensors and colors.
> > > 6. A discussion on the new experiment on dynamic pruning has been added to the experiments.
> > > 7. A few spelling mistakes also have been corrected.
> > > 8. Supplementary material now contains the code for FACS and also FBS for squeezing.
> > >
> > > Kindly go through the paper, and please let us know that it addressed all of your concerns.

---

### Official Review · Reviewer_9T37 · 2022-10-24

**Confidence:** 4
**Correctness:** 3
**Technical Novelty And Significance:** 2
**Empirical Novelty And Significance:** 2
**Recommendation:** 5

**Clarity, Quality, Novelty And Reproducibility:**

This paper is well-written and hence has good clarity and reproducibility.

The technical novelty is limited, as shown in [Weaknesses].

**Strength And Weaknesses:**

[Strengths]
+ This paper is well-written and easy to follow. Most operations are illustrated with corresponding figures and formulas.
+ The manuscript conducts several experiments.

[Weaknesses]
- As the authors sensed, the reader is intuitively aware that the proposed FACS indeed similar to the existing SE-Net [8] and CBAM [25]. The FACS employs the squeeze and channel reduction steps from SE-Net and then excites each channel with the spatial attention as CBAM. Therefore, a supported theory to further explain why composing these operations in the FACS is sufficient and what operations cause its benefits are needed to clarify the work's contribution and novelty.


**Summary Of The Paper:**

In order to speed up the channel squeezing task, this paper proposes the Fast Adaptive Channel Squeezing (FACS) module to replace the conventional 1x1 convolution operator used in channel reduction. The key benefit of FACS is that it neither alters the number of parameters nor affects the accuracy of a given network. Experiments show that FACS reduces FLOPs yet keeps model performance.

**Summary Of The Review:**

The primary concern of this paper is its technical novelty. Several experiments support the proposed FACS, yet no theoretical explanation or invention of new operations. Therefore, it is doubtful if the proposed method could reach the high standard bar of ICLR.

---

> ### Author Response · Authors · 2022-11-15
> **FACS is not similar to SE and CBAM. All of them have different purpose.**
>
> Dear reviewer, thank you for reading our paper. We are encouraged that you found our paper well experimented and well illustrated. A major concern of yours has been novelty which we clarify here:
>
> **1. As the authors sensed, the reader is intuitively aware that the proposed FACS indeed similar to the existing SE-Net [8] and CBAM [25]. The FACS employs the squeeze and channel reduction steps from SE-Net and then excites each channel with the spatial attention as CBAM.**
>
> In Sec.3.6, we describe potential overlap and the major differences of FACS with existing approaches such as SENet and  CBAM. We mention key differences below:
>
> 1. The primary difference is the utility. FACS targets a novel problem of fast channel squeezing by reducing FLOPs requirements while maintaining the parameters and not sacrificing accuracy of the network. Whereas SENet and CBAM improves network accuracy by incorporating more parameters into the network and slightly increased FLOP requirement. This differentiate FACS from SENEt and CBAM at the first place.
>
> 2. Even structurally, FACS is extremely different from both SENet and CBAM. For example, SENet add two new convolution layers which are connected sequentially with ReLU and sigmoidal activation. One squeezes by a factor of 16 while another expands by the same factor and later input and the resulting vector are multiplied. On the other hand, FACS reuses the parameters of the already existing squeeze layer and does not add new layers. Moreover, SENet or CBAM does not have aprovision of fusion in channel dimension.
>
> 3. Most importantly, SE/ CBAM can function together as demonstrated in experiments 4.2.1.
>
> 4. Our intention to keep the paragraph was to strictly avoid confusions to the reader, since FACS uses common terms such as global pooling and squeezing. However, these approaches are orthogonal to FACS.
>
> Hence, we respectfully say that question of novelty for FACS is not justified because SENet and CBAM are very different approaches for very different applications. This also has been described in Sec 3.6.
>
>
> **2. Therefore, a supported theory to further explain why composing these operations in the FACS is sufficient and what operations cause its benefits are needed to clarify the work's contribution and novelty.**
>
> We apologise that the text was not clear to you much. However, instead of describing a unified theory, we described the theoretical functioning and motivations of global context extraction, cross channel information blending and channel fusion in their respective sections. However, we now have improved the text and would like to share with you some theoretical aspects.
>
> The first and foremost step of FACS is to gather global spatial context which is inspired by the fact that it is most frequently employed at the end of the classification networks to aggregate the knowledge of all spatial locations of the final feature map into a 1D descriptor. It is also utilized by SENet, CBAM, and many other works to compute global spatial statistics. Global pooling essentially can work as aggregating a whole channel and it is operated upon by a convolution layer with sigmoidal activations. The sigmoidal activations essentially are used for aggregating knowledge across chunk of k-channels at a given spatial location in a feature map. Then based on the sigmoidal activation, either max or avg aggregation is done using threshold or weighted fusion technique.
>
> In the overall operation, the dominant channels in a particular chunk are forwarded to the next step. In max pooling only one of the channel goes out while in the AVG pooling, all the channels in the chunk are summed. Interestingly, each spatial location has flexibility to choose among different aggregation operation i.e. Max or AVG. Hence, at one location, any one of the k-channels can make it to the output in max pooling while at another location, some other channel makes it to the output. This flexibility or degree of freedom to choose dynamically add a high level of non-linearity into the network which helps achieving FACS slightly better accuracy even at 25\% lower FLOP requirements. These improvements brings huge training and inference speed-ups as shown in the experiment section-4.2.1 for different hardware.

---

> > ### Author Response · Authors · 2022-11-15
> > **Contd...**
> >
> > **3. Several experiments support the proposed FACS, yet no theoretical explanation or invention of new operations. Therefore, it is doubtful if the proposed method could reach the high standard bar of ICLR.**
> >
> > Thank you for recognizing the extensive experimentation. We respectfully say that the problem statement is both new and of practical importance, and has not been addressed previously. Our FACS presented in the paper is described with ample figures and mathematics and acheives significant results. We also have reported analysis for different hardwares i.e. 5 different GPU platforms including heavy duty desktop grade to low power edge computing devices which itself quite challenging to report. We also have provided insight to internal representations of learnt by FACS and shown that they are much better. Being a very new concept for new problem, we strongly  believe and respectfully say that the paper meets the high standard bar of ICLR.
> >
> > However, as pointed by you that you faced difficulty in the theoretical explanation, we have updated the text and upon your next feedback, the paper will be improved further and then will be uploaded.
> >
> > Hopefully, we have satisfactorily answered your major concern. Please let us know in case you want to know more details. Conclusively, we say that FACS is novel and targets a problem which is of practical important but is unattended previously. For this reason, it may be difficult to understand the motive of FACS at a first glance.
> >
> > Finally, we look forward to have a discussion with you. Please let us know in case if you have any further questions.

---

> > > ### Author Response · Authors · 2022-11-17
> > > **New draft uploaded**
> > >
> > > Dear reviewer,
> > >
> > > We have updated the paper. Due to change in citation style, we had to make adjustments to the text.  The changes are as given below.
> > >
> > > 1. Introduction slightly modified. Now mentioning that channel squeezing is important.
> > > 2. Related work updated. Now contains discussion on dynamic pruning.
> > > 3. A theoretical viewpoint added in Sec. 3.4 (new draft). To accommodate, the effect on memory and computational complexity example has been moved to appendix. Also the FLOP equations from the supplementary material has been moved to appendix in the paper for ease of reference.
> > > 4. "Potential similarity.." Sec 3.6 in the initial submission has been moved in related work section to save space.
> > > 5. Figure-4 updated with detailed dimensions of the tensors and colors.
> > > 6. A discussion on the new experiment on dynamic pruning has been added to the experiments.
> > > 7. A few spelling mistakes also have been corrected.
> > > 8. Supplementary material now contains the code for FACS and also FBS for squeezing.
> > >
> > > Kindly go through the paper, and please let us know that it addressed all of your concerns.

---

### Official Review · Reviewer_LDvY · 2022-10-25

**Confidence:** 4
**Correctness:** 3
**Technical Novelty And Significance:** 3
**Empirical Novelty And Significance:** 2
**Recommendation:** 5

**Clarity, Quality, Novelty And Reproducibility:**

## Quality

This paper provides an interesting idea about squeezing the channels at run-time in a dynamic fashion. This dynamic operation helps the networks to preserve its full capacity but can use a smaller number of FLOPs for the given task. The evaluation of this proposed method is concerning, it only makes a comparison to baseline models, this paper has ignored the entire field of dynamic pruning. These flaws in evaluation have a great impact of the quality of this paper.

## Clarity

- This paper has used some non-standard reference styles, and also its cross reference style looks a bit abnormal to me.
- Figure 4 is very confusing. I would prefer you label the input and output dimension of each block, and give the block a name and say the weights (eg. the same style in Figure 1) are having these shapes. The illustration is connecting weights together, which is fairly strange.

## Novelty and Reproducibility

It is hard to judge the novelty of the proposed paper without the authors giving me a full explanation of how it is different from other dynamic pruning techniques.

**Strength And Weaknesses:**

## Strength

- The proposed run-time squeezing scheme, although similar to dynamic pruning techniques, is used as a plug-and-play component. This seamless integration can directly improve the runtime performance of many existing CNNs.
- The authors reported latency numbers on a range of hardware platforms.
- The design choices are well justified in Section 4.1

## Weakness

The major weakness of this paper is on its evaluation, it is not clear to me how this proposed method compares to other published methods.

- This paper does not provide a direct comparison to many existing techniques in the field of dynamic pruning [1, 2, 3]. These dynamic channel pruning techniques are closely related to this work.
- In all the results presented, the authors are mainly comparing to baselines. For example, Table 2 mainly compares a modified ResNet50 with FACS-ResNet50. This does not really provide me with a full picture, what if the ResNet50 is modified further to have the same number of FLOPs? What is the top-1 accuracy gap in this case?
- The proposed technique performs a lot better on ‘old-school’ CNNs like VGG and ResNet. On more trimmed model such as MobileNet, the proposed method seems to have diminishing returns. Also models like MobileNet-V3 and EfficientNet might need to be considered.

[1] Dynamic Channel Pruning: Feature Boosting and Suppression

[2] Channel gating neural networks

[3] Boosting the performance of cnn accelerators with dynamic fine-grained channel gating

**Summary Of The Paper:**

This paper presents a run-time channel squeezing technique. The proposed technique mainly works on the convolutional block and can serve as a plug and play component to many CNNs. The proposed methods does not keeps all possible parameters but can reduce the number of FLOPs to enable fast inference on low-power devices.

**Summary Of The Review:**

This paper proposed an intersting idea, the fact that the proposed method can serve as a plug-and-play component is interesting. However, I do not think the authors have presented a detailed comparison to many existing works in this domain, and this generally affects the quality of this paper and makes it hard to gauge the real contribution.

---

> ### Author Response · Authors · 2022-11-15
> **Pruning is entirely orthogonal to FACS.**
>
> Dear Reviewer, Thank you for reading our paper in depth. We are encouraged that you found it interesting to read the paper’s content. Despite the fact, there has been some concerns of yours which we try our best to resolve here. Before doing that, we would like to provide a quick recap of the paper:
>
> **Our goal:** Develop a channel squeezing method which can preserve the representation power of the network while reduce the FLOP requirements and latency.
>
> **What is not our goal:** Channel pruning whether static or dynamic is orthogonal to our method. They might look similar at first glance but are are entirely different in terms of objective and functionality.
>
> Firstly **The proposed methods does not keeps all possible parameters** is not true. FACS keeps all the parameters intact but reduces FLOPs significantly while maintaining accuracy.
>
> **Addressing Weakness**
>
> **1. This paper does not provide a direct comparison to many existing techniques in the field of dynamic pruning [1, 2, 3]. These dynamic channel pruning techniques are closely related to this work:**
>
> **response:** As mentioned in Sec 2, our method is not a pruning method and is significantly different from the the pruning methods, even the recent dynamic pruning method such as [1,2,3]. Our method is computationally efficient channel squeezing operation which to the best of our knowledge is not studied in the literature explictly. The problem is entirely new and of practical importance. Below we differentiate between FACS and pruning by taking example of [1].
>
> **FACS:** Our method consumes a tensor of size $\mathbb{R}^{C \times H \times W}$ and outputs a tensor $\mathbb{R}^{(C/r)  \times H \times W}$, where r is the squeezing factor generally set to 4. Our method reuses the parameters of the channel squeezing layer and does not introduce any parameter overhead, instead reduces the computational requirements, without requiring specialized convolution implementations.
>
> **Dynamic Pruning:** In the pruning methods, a tensor is of size $\mathbb{R}^{C \times H \times W}$ and still outputs a tensor $\mathbb{R}^{C  \times H \times W}$, but some channels have zero values. This tensor when fed to a convolution, the channels in the input which are turned off or have zero values are not used in the computations (multiply, addition), thus saving the computations during the inference time at the cost of accuracy reduction. For example, the dynamic pruning method [1] performs the same operation. The major difference with such approaches is that they also incorporate new parameters to decide which channels to keep dynamically.
>
> However, unfortunately, pruning can not be applied for channel squeezing as it would lead to change in the overall architecture. Hence FACS and pruning methods are quite orthogonal. The first glance similarity comes from the fact that FACS used terms such as global pooling. FACS is just an architectural modification to save computations by performing the squeezing operation in a fundamentally different manner. Pruning is still applicable to the FACS enhanced networks.
>
> As mentioned that FACS and dynamic pruning can not complement each other because of their fundamentally different operations, there is no way both of them can be compared. However, we push ourself based on your concern, and we are only left with a choice to slightly modify any dynamic pruning technique, based on your feedback, we choose [1] which is the recent state-of-the-art published at ICLR in dynamic pruning, called FBS. We slightly modify the procedure to perform channel squeezing. This approach while pruning, keeps top-k channels and turn remaining to zero, based on a 1-D descriptor of $C$ numbers if input channels were $C$. In the squeezing operation, we are interested in reducing the channel count by a factor of $r$, therefore we keep the top-k channels, where k equals to $C/r$. Now the FBS method outputs $\mathbb{R}^{(C/r)  \times H \times W}$ instead of $\mathbb{R}^{C  \times H \times W}$ which was its default behavior.
>
> While training, we choose ResNet-50, because these models are of much interest to the community instead of smaller ones like ResNet-18 on which FBS is evaluated. During training, we see convergence issues. This happens because when keeping only top-k channels, the intermediate channels are dropped based on the input but it is a random behavior depending on the channel saliency [1] (different for different input). When this tensor is passed to the subsequent convolution, due to dynamically changing position of channel in the output, the convolution struggles to learn the relation between the channels, because channel indexing is lost during choosing top-k.
>
> Secondly, FBS requires an additional $1\times1$ convolution to compute channel saliency which has the weights $\mathbb{R}^{C  \times C}$ in contrast to $\mathbb{R}^{C/r  \times C}$ of FACS. It leads to roughly 48% rise in the parameters i.e. from $25$M to $37$M, which is not desirable.

---

> > ### Author Response · Authors · 2022-11-15
> > **Contd...**
> >
> > **2. In all the results presented, the authors are mainly comparing to baselines. For example, Table 2 mainly compares a modified ResNet50 with FACS-ResNet50. This does not really provide me with a full picture, what if the ResNet50 is modified further to have the same number of FLOPs? What is the top-1 accuracy gap in this case?**
> >
> > As mentioned previously, that FACS is a novel concept to perform channel squeezing in popular bottlenecks, and such kind of work we could not find in literature. Since, we are interested in reducing the FLOPs, the only choice for comparison is the baseline ResNet or any other network chosen.
> >
> > Based on your feedback of dynamic pruning, we tried to adapt one of the pruning method for our problem statement but unfortunately they are not compatible.
> >
> > Moreover, baseline ResNet is not modified, it is exactly the same architecture as presented in the original paper. As we are interested in reducing FLOPs, we were curious to play with reduction ratio which is set to $4$ by default for ResNet. We perform this ablation by changing the ratio to $8$, and find that FACS is still superior by in terms of accuracy and FLOPs
> >
> > Now as you say that *what if ResNet is modified to have same number of FLOPs*, then comparison will have to be made with the FACS variant of the modified baseline, otherwise it would not be fair. Infact the same experiment is there in the Table 2 (mentioned previously), where we change the reduction ratio to 8.
> >
> > Although our goal is not to improve accuracy but FLOPs (Sec 2.2), however FACS also improves the accuracy instead of reducing it unlike dynamic pruning method FBS, which is a huge achievement.
> >
> > **3. The proposed technique performs a lot better on ‘old-school’ CNNs like VGG and ResNet. On more trimmed model such as MobileNet, the proposed method seems to have diminishing returns. Also models like MobileNet-V3 and EfficientNet might need to be considered.**
> >
> > VGG and ResNet despite being older but are still dominant to form newer benchmarks, in robotics especially due to their simplicity. VGG style is till popular due to its low inference time, lower branching. Hence we choose these networks for our target while they also have high representation power.
> >
> > Goal of demonstrating MobileNet-v2 is to show FACS compatibility with depthwise separable convolutions, but these models already have lower FLOPs. Definitely FACS improve FLOPs performance by 21% at the cost of slightly reduced accuracy, is still acceptable considering the accuracy drop on pruning methods. Interestingly, fully-connected variant of MobileNet-v2 with FACS achieves accuracy improvements but at lower FLOPs.
> >
> > MobileNet-v3 and EfficientNet are also based on depthwise separable convolution and have similar architecture, therefore FACS can seamlessly integrated with them. In the paper, we have shown the most popular ones in terms of real time applications with high representation power.
> >
> > **Clarity**
> >
> > We have updated the reference style and also the figures. We will soon upload the new draft after a bit of more polishing and your newer feedbacks.
> >
> > **Review Summary:  The evaluation of this proposed method is concerning, it only makes a comparison to baseline models, this paper has ignored the entire field of dynamic pruning. It is hard to judge the novelty of the proposed paper without the authors giving me a full explanation of how it is different from other dynamic pruning techniques.**
> >
> > Hopefully, we have explained satisfactorily all of your concerns. We also explained how FACS tackles an entirely novel and practically important problem which makes significant improvements in runtime, FLOPs as demonstrated by our extensive experiments on different hardware, bigger and smaller networks, demonstrating learned intermediate network representations.
> >
> > Finally, we look forward to have a discussion with you on further improving the paper.

---

> > > ### Author Response · Authors · 2022-11-15
> > > **Code link for the FBS in squeezing**
> > >
> > > We wanted to share the code for FBS adpated for squeezing but we do not see any appropriate platform to do that anonymously.

---

> > > > ### Author Response · Authors · 2022-11-17
> > > > **New Draft of the paper Uploaded**
> > > >
> > > > Dear reviewer,
> > > >
> > > > We have updated the paper. Due to change in citation style, we had to make adjustments to the text.  The changes are as given below.
> > > >
> > > > 1. Introduction slightly modified. Now mentioning that channel squeezing is important.
> > > > 2. Related work updated. Now contains discussion on dynamic pruning.
> > > > 3. A theoretical viewpoint added in Sec. 3.4 (new draft). To accommodate, the effect on memory and computational complexity example has been moved to appendix. Also the FLOP equations from the supplementary material has been moved to appendix in the paper for ease of reference.
> > > > 4. "Potential similarity.." Sec 3.6 in the initial submission has been moved in related work section to save space.
> > > > 5. Figure-4 updated with detailed dimensions of the tensors and colors.
> > > > 6. A discussion on the new experiment on dynamic pruning has been added to the experiments.
> > > > 7. A few spelling mistakes also have been corrected.
> > > > 8. Supplementary material now contains the code for FACS and also FBS for squeezing.
> > > >
> > > > Kindly go through the paper, and please let us know that it addressed all of your concerns.

---

> > > > > ### Comment · Reviewer_LDvY · 2022-11-18
> > > > > **Completely orthogonal is a bold claim**
> > > > >
> > > > > First, I apologize that in my original review 'The proposed methods does not keeps all possible parameters', I made a wirting mistake, what I meant is the propsoed method does keep all possible paramters. Sorry for the confusion on that.
> > > > >
> > > > > However, I do not think the proposed method is 'orthogonal' to run-time pruning. The purpose of both methods are preserving the representation power of the network while reduce the FLOP requirements depending on run-time information. If you look at your Figure 4, the FACS-ResNet block combined, this is the same as picking the C/R channels to do the expensive computation and then put it back to the origianl dimension. This is not too different from dynamic pruning in my opinion.
> > > > >
> > > > > >However, unfortunately, pruning can not be applied for channel squeezing as it would lead to change in the overall architecture
> > > > > >
> > > > >
> > > > > I am not asking you to put pruning on top of your channel squeezing. I am asking for a direct comparison with these dynamic pruning methods. Surely there is a convergence issue because these methods are designed for fine-tuning (prune-after-train)? Why not just provide a table and compare to [1, 2, 3]? You can maybe address some of your advantages (eg. theoretical speedup is reachable on GPUs, less parameters, no need to fine-tune), but you also need to show the full picture of how these FLOP reduced, dynamic pruned networks compare with you in terms of accuracy?
> > > > >
> > > > > >Goal of demonstrating MobileNet-v2 is to show FACS compatibility with depthwise separable convolutions, but these models already have lower FLOPs. Definitely FACS improve FLOPs performance by 21% at the cost of slightly reduced accuracy, is still acceptable considering the accuracy drop on pruning methods. Interestingly, fully-connected variant of MobileNet-v2 with FACS achieves accuracy improvements but at lower FLOPs.
> > > > > >
> > > > >
> > > > > I have to remind the authors that 0.67%-1.01% accuracy drop is not 'slightly reduced' for these models. Or the other way of looking at it is your method can only increase the accuracy by ~0.1% on these models with almost the same number of FLOPs. If I slightly trim my MobileNet-v2 to use 21% less FLOPs, are you sure this trimmed model will have a large accuracy gap compared to your proposed method?
> > > > >
> > > > > >MobileNet-v3 and EfficientNet are also based on depthwise separable convolution and have similar architecture, therefore FACS can seamlessly integrated with them. In the paper, we have shown the most popular ones in terms of real time applications with high representation power.
> > > > > >
> > > > >
> > > > > The performance gain obtained with proposed method is generally small, and I suspect this gain becomes even smaller if one uses more advanced architectures (eg. MobileNet-V3), because there is now less fat to trim. In this case, I think more advanced architectures are almost necessary for this work.
> > > > >
> > > > > I increased my score to a four since the authors fixed many writing issues. However, my major concerns on evaluation and comparison with related work are still not fully addressed. One honest suggestion, there is no point to iterate on the novelty claim, what reviewers (or at least me) ask for is to position your work at the correct place. I do not think making bold claims such as this is completely orthogonal would help.

---

### Author Response · Authors · 2022-11-16
**FACS targets a new problem in a novel way by leveraging existing fundamental components. The practical importance of the problem is huge.**

To all the reviewers:

Dear reviewers,

1x1 convolution is mainly used as channel squeezing in deep networks despite adding non-linearity. However, there number in a network is substantial i.e. ResNet-50 for instance, consists of 16 such layers which form 33% of total layers and 25% (1.05B/4.12B) of total FLOPs or computations.

As ResNet, VGG style networks are still dominant in both academia and industry,  due to their architectural simplicity, customizability, and high representation in contrast to newer complex networks, the preceding observations compel us to ask a naturally arising question: can computations in 1 × 1 layers be reduced without sacrificing network accuracy or parameters ? If so, the inference of such networks can be significantly accelerated on state-of-the-art edge devices, benefiting a whole spectrum of applications such as autonomous driving, robotics, and so on.

In this paper, we contribute by showing that indeed it is possible by examining the functioning of squeeze layer through the lens of computational complexity. To this end, we propose a novel “fast adaptive channel squeezing” (FACS) module that transforms a feature map $X \in \mathbb{R}^{C \times H \times W}$ to another feature map $Y \in \mathbb{R}^{(C/r) \times H \times W}$.


It is entirely different from pruning methods because of the problem under consideration, also mentioned in the response to **Reviewer LDvY**. FACS employs  global pooling , convolutions which are  very fundamental operations and are also used in large spectrum of networks. However there combination is novel in FACS for the novel problem, as mentioned in Sec. 3.  Further, our channel fusion strategy is entirely novel, and its evidence in this context does not exist in recent literature. Adaptive Max and average fusion, threshold fusion and weighted fusion are the novel component.

Empirically, FACS speeds up networks by 25% without loss of accuracy, even improving it for high representation power networks, as shown in Table-2. Moreover, the throughput and latency gains are huge as shown in Table-3 which are beneficial for fast inference on edge devices, shown in the Table-3 for 5 different GPU devices.

Similar works such as feature boosting and smoothing in dynamic pruning, ICLR 2019 appears simillar because of global pooling operation bt there are several other differences, as demonstrated in response to reviewer LDvY.  Pruning can not complement  FACS. Moreover, SENet, CBAM are methods to improve accuracy of a network by adding new layers, adding more parameters, while FACS does not add new layers, maintains the non-linearity  but improves latency, reduces FLOPs, improves throughput.

Hence, we respectfully say that FACS does not suffer with the novelty constraint. Also, extensive experimentation quantitative, qualitative via GradCAM suggests that FACS is indeed powerful.

Thank you

---

### Author Response · Authors · 2022-11-17
**Rebuttal acknowledgement**

Dear reviewers,

As the deadline is approaching, we kindly request you to go through our rebuttals and suggest additional questions or updates in the paper. Currently, we have revised our paper based on the feedback of all the reviewers, clarified the novelty issue and added theoretical viewpoint to FACS. Also a few new codes has been added in the supplementary material.

Kindly acknowledge the discussion.

Thank you

---

### Decision · Program_Chairs · 2023-01-20

**Decision:**

Reject

**Justification For Why Not Higher Score:**


- Reviewer LDvY also points out that a relevant baseline to compare against would be a model of the same architecture family in which the capacity has been reduced so as to match FACS in terms of FLOPs. The authors counter that FACS is the only approach to perform efficient channel squeezing, and since they are interested in reducing the FLOPs, the only choice for comparison is the unmodified network architecture. They point to Table 2 for experiments on ResNet-50 where the reduction ratio has been set to 8 rather than 4 and which shows that FACS still comes out ahead. I agree with Reviewer LDvY, and from my perspective the authors' response does not address this point directly: Table 2 compares models while controlling for capacity in the form of number of parameters, whereas Reviewer LDvY asks about controlling for FLOPs. The ResNet-50 with a reduction ratio of 8 is not directly comparable to FACS-ResNet-50 in that regard because they require 1.85B and 3.18B FLOPs, respectively. The question of how much of an accuracy drop one would suffer by adjusting the numbers of channels in a ResNet-50 architecture so as to match the FLOPs in FACS-ResNet-50 is relevant, and it still stands.
- Reviewers 9T37 and Ywmp note the similarity in architectures between FACS and the existing SENet and CBAM architectures. The SE block performs global average pooling followed by a single-layer (dimensionality-preserving) nonlinear (ReLU) projection and multiplies the result channel-wise with the (unmodified) input. FACS performs global average pooling followed by a single-layer (dimensionality-reducing) nonlinear (sigmoid) projection and multiplies the result with the (channel-wise max-and/or-average-pooled) input. CBAM's channel attention module performs channel-wise max-pooling and average-pooling, passes both independently through the same MLP, then passes the summed outputs through a (sigmoid) nonlinearity and multiplies the result with the (unmodified) input. From Reviewer Ywmp's perspective, the most notable difference resides in the channel fusion strategy, and they would like to see a better characterization of this design choice. The authors reply that while all three share architectural elements, FACS' purpose is different, as it targets efficient channel squeezing, and it results in a network architecture that is different because it is parameter-preserving, unlike SENet and CBAM. They also provide additional experiments characterizing the design choices for the fusion mechanism. Reviewers 9T37 and Ywmp remain unconvinced regarding the submission's technical novelty. On this point, both the authors' and the reviewers' arguments have merit. The authors are right to point out that FACS tackles a new problem and that this constitutes a novel contribution, even if the mechanism through which it approaches the problem shares architectural elements with existing work. However, I understand the reviewers' unease about the way in which FACS is introduced and the authors' insistence on presenting it as a completely new approach. _(On that note, I should clarify that after reaching out to Reviewer LDvY regarding their comment on novelty quoted by the authors, Reviewer LDvY agrees with the other reviewers; what they were suggesting is that the authors simply acknowledge the similarities to previous work and focus on framing their work in that context rather than attempt to convince reviewers that their proposed approach is highly novel.)_ I think it would also be factually correct to present FACS as a channel squeezing approach which replaces the bottleneck with a SENet-like block and pairs it with a channel fusion mechanism to enable channel reduction (which is unaddressed in SENet). It would also be factually correct to say that FACS and CBAM share a design element of channel-wise average and max pooling, although FACS extends the concept to enable channel reduction (which again is unaddressed in CBAM). The authors should be more transparent on this in their discussion of related work.


Ultimately, I believe the reviewer and author disagreement on baselines, comparison against related work, and framing of the proposed approach stems from differing viewpoints rather than a lack of engagement with or understanding of the submission. Since the reviewers remain unconvinced by the authors' arguments, the submission does not quite meet the bar for acceptance.

**Justification For Why Not Lower Score:**

N/A

**Metareview: Summary, Strengths And Weaknesses:**

The submission introduces an approach called Fast Adaptive Channel Squeezing (FACS) which aims to carry out the channel squeezing operation in convolutional neural networks in a computationally efficient manner and without affecting capacity and accuracy. FACS operates in three stages on an input of shape [C, H, W]:

1. a global average pooling operation is applied to obtain a 1D descriptor of shape [C, 1, 1];
2. information is blended across channels through a linear mapping followed by an elementwise sigmoid nonlinearity, resulting in a 1D fusion probability vector of shape [C / R, 1, 1]; and
3. the input channels are partitioned into chunks of R channels and squeezed through a fusion operation (which operates independently for each spatial position) using the fusion probability vector, resulting in an output of shape [C / R, H, W].

Two fusion strategies are proposed:

1. threshold based function computes either the max or average over a given chunk of channels (depending on whether the corresponding fusion probability is below or above some threshold), and
2. weighted fusion computes a convex combination of the max and average using the corresponding fusion probability).

Empirical evaluation on three architecture families (VGG, ResNet, MobileNet-v2) and on three datasets (ImageNet, CIFAR-10, CIFAR-100) and FACS is shown to be around 23% faster (in terms of FLOPs) while offering a marginal accuracy improvement in most cases . Ablation studies are also performed to assess the effect of the fusion activation function, the interaction of FACS with batch normalization, the effect of the fusion threshold, and the impact of the channel fusion strategy. Finally, transfer learning and GradCAM visualizations are presented.

Reviewers note that the proposed approach is effective at reducing the computational cost while maintaining or improving performance (Ywmp), that it is easily applied to many CNN architecture families (LDvY), and that many experiments are presented to validate it (LDvY, 9T37). The main reviewer concerns revolve around baselines, comparisons against existing work, and technical novelty:

- Reviewer LDvY points out that the submission does not compare against dynamic pruning approaches, which in their opinion are closely related. The authors respond by clarifying that FACS' objective is to develop a channel squeezing method which can preserve model capacity while reducing computational cost, whereas dynamic channel pruning seeks to save on computation at inference time at the cost of accuracy reduction. They argue that pruning cannot be applied to channel squeezing, as it incorporates new parameters and modifies the network architecture, and that FACS and pruning are therefore orthogonal and cannot be compared. They mention an attempt to train a modified FBS pruning approach which discards channels not in the top-k instead of zeroing them out and report convergence issues and an increase in model parameters. Reviewer LDvY disagrees with the authors: from their perspective, both FACS and run-time pruning seek to preserve model capacity while reducing the computational cost, and pruning approaches do not necessarily need to be used to perform channel squeezing in order for them to be directly comparable. Both sides continue to argue their points further, but at this stage I should say that their thoughts have been made clear, and I agree with Reviewer LDvY. In the reviewer discussion, Reviewers 9T37 and Ywmp also agree. I see the importance of reducing computational requirements and maintaining model capacity, but I don't see why this necessarily has to be done through improved channel squeezing. FACS may be the first approach to more computationally efficient channel squeezing, but it's not the first work which looks into reducing the computational cost of a network while maintaining good accuracy. As such, it should be possible to compare model accuracies while controlling for FLOPs, and if as the authors claim pruning makes too large an accuracy sacrifice for the computational savings, FACS would be clearly shown to be superior in that regard. The analogy to SENet and its evaluation protocol put forward by the authors does not quite hold, either: SENet was used by the winning entry in the ILSVRC 2017 classification competition, and as a result was compared against a plethora of competing approaches (see Table 8 in the SENet paper).

(continued in "Justification For Why Not Higher Score")

**Summary Of Ac-Reviewer Meeting:**

N/A